# Compaction Properties of Particulate Proteins in Binary Powder Mixtures with Common Excipients

**DOI:** 10.3390/pharmaceutics16010019

**Published:** 2023-12-22

**Authors:** Else Holmfred, Cosima Hirschberg, Jukka Rantanen

**Affiliations:** Department of Pharmacy, Faculty of Health and Medical Sciences, University of Copenhagen, 2100 Copenhagen, Denmark

**Keywords:** protein, pharmaceutical, tableting, powder properties, compaction

## Abstract

The increasing interest in protein- and peptide-based oral pharmaceuticals has culminated in the first protein-based products for oral delivery becoming commercially available. This study investigates the compaction properties of proteins in binary mixtures with common excipients up to 30% (*w*/*w*) of particulate protein. Two model proteins, lysozyme and bovine serum albumin, were compacted with either microcrystalline cellulose, spray-dried lactose monohydrate, or calcium hydrogen phosphate dihydrate at two different compaction pressures. Compared to the compacted pure materials, a significant increase in the tensile strength of the compacts was observed for the binary blends containing lysozyme together with the brittle excipients. This could be attributed to the increased bonding forces between the particles in the blend compared to the pure materials. The use of bovine serum albumin with a larger particle size resulted in a decrease in tensile strength for all the compacts. The change in the tensile strength with an increasing protein content was non-linear for both proteins. This work highlights the importance of considering the particulate properties of protein powders and that protein-based compacts can be designed with similar principles as small-molecules in terms of their mechanical tablet properties.

## 1. Introduction

Oral delivery of proteins has been attempted for almost 100 years, with only a few drugs reaching the market [1,2]. Currently, the interest in oral drug products containing peptides and proteins is increasing [3,4], and culminated in 2019 when the oral glucagon-like peptide 1 (GLP-1) agonist, semaglutide (Rybelsus^®^), was approved by the Food and Drug Administration [3,5]. The oral route of administration is the most preferred due to its simplicity and convenience for the patients when contrasted with injectables [6,7]. Compared to small molecules, the structure of proteins has several complicating characteristics, e.g., the amino acid sequence and the unique folding that determine their function [7,8]. Although the number of approved medicines containing proteins is increasing, the delivery of proteins remains challenging given the low absorption and related poor bioavailability of peptides and proteins [1,3,9,10].

There is a clear interest in the development of oral protein formulations; however, the manufacturing of tablets involves stress factors like pressure, temperature, and exposure to environmental conditions that potentially can affect the protein structure. The unique protein structure should be maintained during processing, and potential structural changes must affect the functionality of the proteins minimally [7,8]. This has been investigated in several studies in which the activity and structural integrity of proteins after compaction have been analyzed [10,11,12,13,14]. The sensitive protein structure [10,15,16] consequently forces most protein products to be formulated as injectable drugs, either liquid formulations ready for injection or freeze-dried powder formulations, reconstituted right before administration to the patient.

Few studies have investigated the consequence of the compaction of proteins [10,12,14,17], but limited attention has been paid to the particulate properties of peptides and proteins and the related secondary processing. Klukkert et al. (2015) investigated the conformational changes in trypsin due to compaction, where a decrease in the enzymatic activity of trypsin was found with an increasing compaction pressure [10]. Controlling and pre-conditioning the relative humidity of the protein is potentially a critical parameter for protein stability and the level of aggregation before and during compaction. Wei et al. (2019) reported the effect of compaction on the stability of pure lysozyme and BSA and found that both proteins were highly plastic when compressed individually and showed excellent tabletability. The enzymatic activity of lysozyme was unchanged up to a compaction pressure of 300 MPa. BSA was seen to be more sensitive to compaction pressure at a small particle size, and a high compaction pressure showed upon compression a tendency for aggregation in the tablet. The authors found lysozyme and BSA stable after compaction, even when pre-conditioning at elevated humidities [12]. On the contrary, Lu et al. (2021) described how changes in relative humidity storage conditions induced conformational changes and the aggregation of immunoglobin prior to compaction. Immunoglobin was pre-conditioned at 32%, 52%, 75%, or 93% for 1 month, and the level of aggregation increased with increasing relative humidity. The immunoglobin binding activity was unaffected by the aggregation induced by changes in the relative humidity, while compaction pressures (50–350 MPa) did not induce aggregation but changed the binding activity significantly [17]. The studies from Klukkert et al. (2015), Lu et al. (2021), and Wei et al. (2019) focused on the protein stability and aggregation prior to or after compaction but did not investigate the protein as a particulate matter [10,12,17]. A recent study by Pedersen et al. (2023) described the tabletability properties of an oral drug formulation containing insulin, however, placing the focus on improving the often poorly tabletability of formulations containing high doses of permeation enhancers [14]. As permeation enhancers are often expensive, the authors replaced permeation enhancers with maltitol and added maltitol to the spray-drying process of insulin. The tensile strength of the tablets was significantly improved with the use of small-sized insulin–maltitol particles (2.16 µm, Dx50) compared to large-sized insulin–maltitol particles (4.89 µm, Dx50). Incorporating small-sized insulin–maltitol particles into the formulation did not influence the drug release of insulin from the formulation. Focusing mainly on improving the tabletability parameters, the stability of insulin or the binding activity was not addressed [14].

The particulate properties of both the drug and the excipients are of critical importance when designing a solid dosage form. This is well documented in the literature for small-molecule-based oral products and has recently been summarized in the Manufacturing Classification System [18]. Also, extensive work on the compaction of binary powder blends and the impact of different particulate factors on the percolation threshold of a binary blend has been the focus of research [19,20,21,22,23]. Identifying the concentration and the particulate properties of an active ingredient that impacts the downstream processability is crucial to ensure a robust production process.

The aim of this work was therefore to investigate the impact of increasing concentrations of two model proteins, lysozyme and bovine serum albumin (BSA), in binary mixtures containing three different model excipients (spray-dried lactose monohydrate (LacMH), microcrystalline cellulose (MCC), and calcium hydrogen phosphate dihydrate (CaHP)). The excipients used in this work were chosen based on their different particulate properties, leading to diverse behavior during the tableting process, and commercial availability. MCC is one of the most commonly used excipients in direct compression due to its high plasticity and ability to form tablets with high tensile strength [24,25,26]. LacMH and CaHP are less plastic than MCC [25,27], although they are also commonly used excipients in tablets.

## 2. Materials and Methods

### 2.1. Materials

The lysozyme from chicken egg white and bovine serum albumin (BSA) (Sigma-Aldrich, Schnelldorf, Germany), MCC (Avicel PH-102, FMC, Cork, Ireland), CaHP (EMCOMPRESS Premium, JRS Pharm, Rosenberg, Germany), and LacMH (FlowLac 100SD, Meggle, Wasserburg, Germany) were all of analytical grade. The magnesium stearate and potassium carbonate were purchased from Sigma-Aldrich, Schnelldorf, Germany.

### 2.2. Particle Size Distribution

A laser diffraction particle sizer with a Scirocco dry powder feeder (Mastersizer 2000, Malvern Instruments, Worcestershire, UK) was used to analyze the particle size distribution of the protein powders (n = 3).

### 2.3. X-ray Powder Diffraction

The X-ray powder diffraction measurements were performed using an X’Pert PANalytical Pro X-ray diffractometer (PANalytical, Almelo, The Netherlands) (CuKα radiation, λ: 1.54187 Å; acceleration voltage: 45kV; current: 40 mA; reflectance mode: 5°–35° 2θ; scan rate: 0.067° 2θ/s; step size: 0.026°). The data were collected and analyzed using the software X’Pert Data Collector version 2.2 (PANalytical, Almelo, The Netherlands).

### 2.4. Dynamic Vapor Sorption

The dynamic vapor sorption profiles of the proteins were analyzed using a VTI-SA+ instrument (TA-Instruments, New Castle, DE, USA). Approx. 15 mg of the powder sample was used (n = 1). The samples were dried inside the VTI-SA+ at 60 °C for either 180 min or until weight equilibrium (less than 0.0010 wt% change in 5 min) was achieved. Afterward, the temperature was adjusted to 25 °C, and the relative humidity (RH) was increased stepwise to 95% RH. Steps of 10% RH were chosen up to 90% RH, followed by one 5% step to 95%RH. The same steps were used for desorption. The sample was equilibrated at each relative humidity for 180 min or until the weight equilibrium was reached.

### 2.5. Sample Preparation

The BSA was gently ground using a mortar and pestle to de-aggregate and ensure the proper mixing behavior. Lysozyme and BSA were mixed with MCC, CaHP, and LacMH at three concentrations: 5, 10, and 30% (*w*/*w*) protein content. The sample size was approximately 10 g. All powders were mixed in a Turbula mixer for 3 min at 32 rpm (Type T2F, System Schatz, Willy A Bachhofen AG, Muttenz, Switzerland). All the blends and protein powders were conditioned in a desiccator at 43% RH (saturated potassium carbonate salt solution) for a minimum of 48 h. This condition was selected based on the dynamic vapor sorption analysis. Prior to compaction, 0.5% (*w*/*w*) magnesium stearate was added to the mixtures and mixed for 30 s in the Turbula mixer.

### 2.6. Compaction

All compactions were performed using a compaction simulator (HB10, Huxley Bertram Engineering Limited, Cambridge, UK). The samples (n = 4) were compacted at two different compaction pressures: 136 MPa and 265 MPa. The chosen compaction pressures reflect the range of compaction pressures used for pre-compression and the main compaction during general tablet manufacturing. Equally, 10 mm flat-faced punches were used for compaction with a compaction speed of 2 mm/s. The maximum load was kept constant for 50 ms. The compaction simulator with Euro B type tooling was set to provide a maximum load of 50 kN with a punch speed of a maximum of 2.5 mm/s. The die filling and compact removal were performed manually.

### 2.7. Properties of Compacts

The compacts were characterized after 24 h of storage at ambient conditions. The mass of the compact (n = 4) was determined using an analytical balance (Mettler AJ150, Mettler Toledo, Columbus, OH, USA), and the height of the compact (n = 4) was determined using a stage (7007, Mitutoyo Deutschland GmbH, Neuss, Germany) equipped with a digimatic indicator (Type IDF-130; Mitutoyo Deutschland GmbH, Neuss, Germany). The compact diameter and the crushing strength (n = 4) were measured using a Dr. Schleuniger Pharmatron Tablet Tester 8M (SOTAX AG, Aesch, Switzerland). The elastic recovery of the compacts was calculated using Equation (1)
(1)$$ER=\frac{h_{out\;}-h_{max}}{h_{max}}$$
where *h_max_* is the thickness of the compact at maximum compaction pressure (mm) and *h_out_* is the out-of-die thickness of the compacts measured after 24 h of relaxation (mm). The tensile strength of the compacts was calculated using Equation (2)
(2)$$\sigma =\frac{2F}{\pi \cdot D\cdot T}$$
where *σ* describes the tensile strength of the compacts calculated, *F* is the breaking force of the compacts (N), *D* is the compact diameter (m), and T is the compact thickness (m). The dependence of the tensile strength and tablet density on the compaction pressure was analyzed.

### 2.8. Data Analysis and Statistics

MATLAB 2022b (Mathworks, Natick, MA, USA) was used to extract the relevant data from the compaction profiles. The graphs were evaluated and plotted in Origin Pro 2016 (OriginLab Corporation, Northampton, MA, USA). Student’s *t*-test (α = 0.05) was used to determine whether the two means differed significantly.

## 3. Results and Discussion

### 3.1. Size Distribution of Bovine Serum Albumin and Lysozyme

In this work, BSA and lysozyme are used as model proteins, focusing on their particulate properties. It should be noted that the protein activity after compaction was not evaluated. Particle size is an essential factor influencing the tableting process and, ultimately, the mechanical properties of the compacted product, e.g., the strength of a tablet. For particles to bind and form a compact, interparticulate forces play a crucial role [28]. Therefore, a large surface area of the materials, corresponding to smaller particle sizes, is usually favorable.

One clear difference between the used model protein powders was particle size and protein size. The average particle size of the lysozyme (d50 value) was 13 ± 0 µm and that of BSA was 226 ± 29 µm (Figure 1A).

The particle size of lysozyme was significantly smaller than that of BSA, and based on the particle size, it can be expected that lysozyme will form stronger compacts. The X-ray powder diffractograms indicated the amorphous nature of both proteins (Figure 1B). The amorphous nature of the proteins is interlinked with the water sorption behavior, where lysozyme and BSA demonstrated an ability to absorb a high amount of water (Figure 1C). A comparable water sorption behavior was observed for lysozyme and BSA up to a relative humidity of 60%. With an increasing relative humidity, BSA absorbed more water than lysozyme. At 95% relative humidity, BSA absorbed 41% (*w*/*w*) water and lysozyme 29% (*w*/*w*).

Properties like the ability to deform due to plastic and/or elastic deformation significantly impact the tabletability of a mixture [26,28]. Wei et al. (2019) investigated the effect of particle size on the compaction behavior of lysozyme and BSA and found that the particle size significantly impacts the tensile strength of compacts made purely from lysozyme, whereas the particle size of BSA had a lower impact on the tensile strength. Wei et al. (2019) further evaluated various standard compaction parameters, including Young’s modulus and the contact hardness of the lysozyme and BSA compacts. Lysozyme and BSA demonstrated low Young’s modulus values and a contact hardness consistent with their amorphous nature. The low values of contact hardness suggested the plastic deformation of lysozyme and BSA during compaction, favored by an increasing interparticle bonding area [12]. However, adding a second component to the powder matrix, such as excipients, increases the complexity of the formulation and the compaction behavior.

### 3.2. Binary Blends Containing Bovine Serum Albumin

Physical mixtures of amorphous BSA powder and three common excipients (LacMH, CaHP, and MCC) were prepared at different concentrations of BSA. The tablets were compacted using a compaction simulator at two different compaction pressures (136 MPa and 260 MPa), demonstrating typical low and high compaction pressures in an industrial setting [29]. The characteristic tablet parameters, such as tensile strength and elastic recovery, were monitored with an increasing BSA content (Figure 2A,B). The BSA protein was also compacted as a pure powder to rationalize the effect of BSA in binary blends. It was observed that pure BSA formed coherent compacts, however, with a low tensile strength of 0.3 and 0.6 MPa at the two compaction pressures (136 and 260 MPa) (Figure 2A).

In contrast to our results, Wei et al. (2019) reported that preconditioned (relative humidity 42%) pure BSA compacts (particle size 250–355 µm) had a tensile strength above 2 MPa at a 200 MPa compaction pressure [12]. However, the BSA protein powder quality and grade might differ significantly between this study and the study performed by Wei et al. (2019), which could explain the observed differences in tensile strength.

The excipients showed a higher tensile strength when compacted individually at low and high compaction pressure compared to the binary mixtures. Compacts made of pure CaHP and LacMH had similar tensile strengths, at 1.2 and 2.2 MPa, at compaction pressures of 136 and 260 MPa, respectively. Pure MCC showed the highest tensile strength upon compaction with 9.5 and 11.5 MPa at compaction pressures of 136 and 260 MPa, respectively. Already low amounts of BSA lowered the tensile strength of the used excipient compacts. At 5 w/w% of BSA, the tensile strength of the binary blends containing CaHP was already significantly (α = 0.05) decreased, independent of the compaction pressure. For LacMH and MCC, the addition of 5 w/w% of BSA did significantly decrease the tensile strength of the binary blends at a 136 MPa compaction pressure. At a high compaction pressure (260 MPa), 10 w/w% BSA was needed to have a significant impact on the tensile strength, showing that the percolation threshold of this binary blend is dependent on the compaction pressure [19]. It should be noted that acceptable compacts with tensile strengths higher than 2 MPa could be compacted from all BSA binary mixtures and BSA-MCC blends at both compaction pressures. The particle size of BSA might affect the compactability of the protein; however, a decrease in the particle size of BSA is not expected to suddenly increase the tensile strength in binary mixtures, but is expected to be a material-related property of the protein powder, as seen for other materials [26].

Additionally, the elastic recovery (out-of-die) was measured for the compacts (Figure 2B). BSA compacted alone showed an elastic recovery of 18 and 22% at the 136 and 260 MPa compaction pressures, respectively. The elastic recovery of BSA exceeded the elastic recoveries of both LacMH and CaHP (7% at the 136 MPa compaction pressure and 11% at the 260 MPa compaction pressure). It was observed that already 5 w/w% of BSA had a significantly increasing effect (α = 0.05) on the elastic recovery. An increase in the BSA weight fraction in the binary blends further increased the elastic recovery. MCC showed an elastic recovery of 27.5% at the 260 MPa compaction pressure, which is higher than the elastic recovery for BSA at that compaction pressure. A significant effect of adding BSA to MCC on the elastic recovery at the 260 MPa compaction pressure was seen after adding 30 w/w% of BSA, indicating that at lower concentrations of BSA, the elastic properties of MCC exceed the effect of BSA. At the 136 MPa compaction pressure, the elastic recovery of both MCC and BSA was around 18%. Adding BSA to MCC showed no effect on the elastic recovery at that compaction pressure.

To the best of our knowledge, no studies have considered the particulate properties of protein binary mixtures, though a few studies have considered the importance of the protein stability upon compaction [10,12,17]. Compacting BSA in binary mixtures of three commonly used excipients was possible, indicating that the general concepts of compacting proteins can be used for the design of oral protein drug delivery. However, the challenges of peptide and protein oral drug delivery, such as poor absorption, gastrointestinal enzymatic digestion, etc. [30] must also be addressed to succeed with oral protein drug delivery. Protein-coating [31], mucus-penetrating agents [32,33], permeation enhancers [14,34], etc. are examples of approaches to overcoming the oral protein formulation challenges, but the strategies are likely a case-to-case-based decision from protein to protein. A common strategy is using permeation enhancers to improve the protein absorption, which may feed new challenges, as permeation enhancers are often poorly compactable [35]. Pedersen et al. (2023) simulated the challenges of compacting poorly compactible permeation enhancers using high concentrations of maltitol in formulations containing insulin. The authors described how the tabletability properties were significantly improved by tailoring the spray-drying process of insulin with maltitol to form small and non-hollow insulin-maltitol particles [14].

### 3.3. Binary Blends Containing Lysozyme

As a second model protein, amorphous lysozyme powder with a smaller mean particle size than BSA was used (Figure 1). Similarly to the BSA experiments, lysozyme was added into three different excipients (LacMH, CaHP, and MCC) at increasing weight fractions (5, 10, and 30 w/w%). The compaction of lysozyme alone resulted in tablets with a tensile strength of 2.0 MPa at a 136 MPa compaction pressure and 2.3 MPa at a 260 MPa compaction pressure (Figure 3A). These results are in good agreement with Wei et al. (2019), where compacted pure lysozyme had a tensile strength of ~2 MPa at a 150 MPa compaction pressure [12].

Comparing pure lysozyme with pure CaHP indicated that at a 136 MPa compaction pressure, the tensile strength of BSA is lower than that of lysozyme (1.1 MPa). At a 260 MPa compaction pressures, the tensile strengths of CaHP and lysozyme are comparable at 2.2 MPa. Interestingly, a significant increase in the tensile strength of binary blends containing CaHP and lysozyme above the tensile strength of the individual ingredients was observed, indicating that a combination of those components has a higher bonding strength within the tablet than the individual materials at a 260 MPa compaction pressure. A similar behavior was seen for binary blends containing LacMH and lysozyme. Surprisingly, this effect was only observed for the higher compaction pressure (260 MPa). This shows that a certain compaction pressure is needed to see this effect of an increased bonding strength.

We suggest that lysozyme in combination with LacMH or CaHP increases the bonding strength via the solid bridge formation or increased mechanical interlocking between the two excipients. Lysozyme could hereby show functionality comparable to common dry binders. However, another possible reason for the increased mechanical properties is the moisture introduced into the blend by the lysozyme, inducing surface sintering at the particle surfaces in contact with the excipient particles. The hypothesis of the increased bonding strength between the two materials (in this case, LacMH or CaHP and lysozyme) is supported by the fact that the tablet density was not affected by the addition of lysozyme at high compaction pressures (Figure 3).

For binary blends containing MCC, this effect was not observed. However, it should be noted that MCC alone forms strong tablets [24,26], and a potential increase in tensile strength could not be detected due to the limitations of the tablet strength tester (individual tablets of MCC/5 w/w% and 10 w/w% lysozyme blends). Compared to BSA, lysozyme showed a higher elastic recovery than CaHP and LacMH and a lower elastic recovery than MCC (Figure 3B). Adding 5 w/w% lysozyme affected only the elastic recovery of the blends containing LacMH. At 30 w/w% lysozyme and 260 MPa compaction, all blends except those containing MCC significantly impacted the elastic recovery. The unexpected increased bonding strength between lysozyme and LacMH or CaHP might result from solid bridge formation or increased mechanical interlocking between protein powder and excipients. Lysozyme could hereby show functionality comparable to common dry binders. However, another possible reason for the increased mechanical properties is the moisture introduced to the blend by lysozyme-inducing surface sintering at the particle surfaces in contact with the excipient particles. The hypothesis of increased bonding strength between the two materials (in this case, LacMH or CaHP and lysozyme) is supported by the fact that adding lysozyme at high compaction pressures does not affect the tablet density. However, to fully understand the impact of particle size of the used model proteins, it would be of benefit to use the same protein in different sizes, as reported by Wei et al. (2019) [12]. Conclusions based on the particle sizes between two different proteins cannot be made, as the intrinsic properties of both proteins vary.

In general, a lower impact on the elastic recovery for blends containing lysozyme was seen compared to BSA (Figure 2B and Figure 3B), and it was possible to compact lysozyme with all three excipients. Wei et al. (2019) reported that the compaction of pure BSA and lysozyme did not cause any detectable changes in the secondary or tertiary structure or the thermal stability [12]. The stability of BSA and lysozyme is not expected to be affected in binary mixtures, but this could be evaluated in a future study.

With this study, we have demonstrated that the particulate properties of binary protein–excipient mixtures, in terms of their tablet properties, share similarities with small molecule-based studies. For developing biological oral tablet products, not only the particulate properties but also the protein stability and activity must be in focus.

## 4. Conclusions

This study showed that compacting up to 30 w/w% of the selected model proteins with commonly used excipients into tablets with a sufficient tensile strength for downstream processing is possible. Especially with MCC, compacts with a tensile strength of more than 4 MPa could be compacted with all the powder blends, which indicates that MCC could be a good choice for tablet-based products containing proteins. With an increasing BSA content, the tensile strength decreased when compared to the tablets with the pure excipient. The change in elastic recovery with an increasing BSA content indicated a strong effect at an already low BSA concentration. The increase in the tensile strength of the mixtures of lysozyme with LacMH and CaHP indicated the formation of stronger bonds between the components in the tablet, suggesting lysozyme acted as a dry binder.

## Figures and Tables

**Figure 1 pharmaceutics-16-00019-f001:**
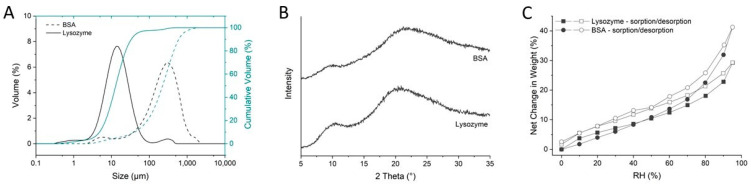
(**A**) Volume-based estimation of the size distribution (black) and cumulative volume-based size distribution (green) of lysozyme (straight line) and bovine serum albumin (BSA) (dashed line) (n = 3, mean). (**B**) X-ray powder diffraction pattern of bovine serum albumin (BSA) and lysozyme (n = 1). (**C**) Vapor sorption (filled symbols) and desorption (open symbols) profiles of lysozyme (squares) and bovine serum albumin (circles) (n = 1).

**Figure 2 pharmaceutics-16-00019-f002:**
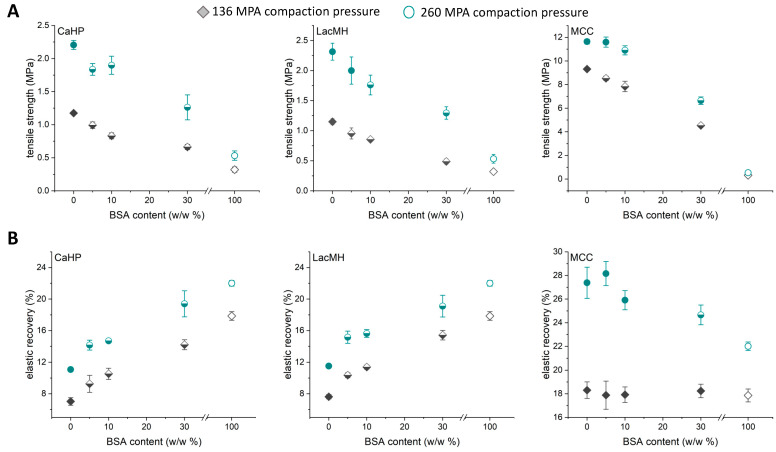
(**A**) Tensile strength of binary blends containing BSA and CaHP, LacMH, and MCC and (**B**) elastic recoveries of binary blends containing BSA and CaHP, LacMH, and MCC. Open symbols: pure protein; filled symbols: pure excipients and binary blends that are not significantly different from pure excipients; half-filled symbols: binary blends that are significantly different from pure excipients (n = 5; mean ± SD). Diamond (♦)—136 MPa compaction pressure; circles (●)—260 MPa compaction pressure.

**Figure 3 pharmaceutics-16-00019-f003:**
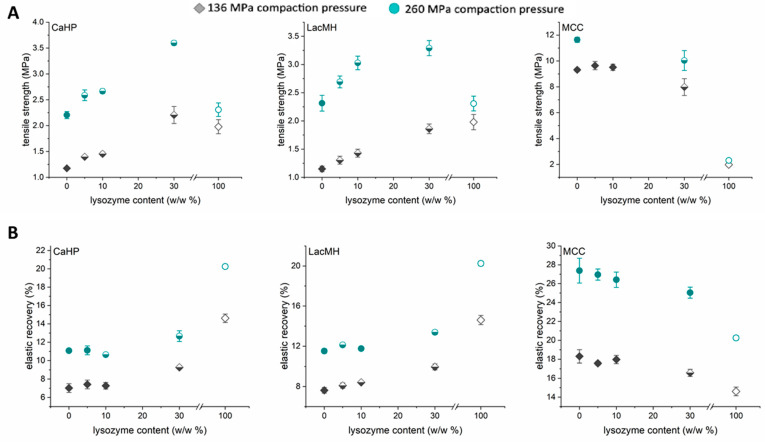
(**A**) Tensile strength of binary blends containing lysozyme and CaHP, LacMH, and MCC. (**B**) Elastic recovers of binary blends containing lysozyme and CaHP, LacMH, and MCC. Open symbols: pure protein; filled symbols: pure excipients and binary blends that are not significantly different from pure excipients; half-filled symbols: binary blends significantly different from pure excipients (n = 5; mean ± SD). Diamond (♦)—136 MPa compaction pressure; circles (●)—260 MPa compaction pressure.

## Data Availability

Data are available upon request.

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
