# Peer review of "Compaction Properties of Particulate Proteins in Binary Powder Mixtures with Common Excipients"

_pharmaceutics, 2023, doi:10.3390/pharmaceutics16010019_

Round 1

Reviewer 1 Report

Comments and Suggestions for Authors

The authors have employed two model  proteins to explore the impact of different tableting excipients on the compaction properties of  binary drug-excipient mixtures. The work is relevant to the increasing interest in oral delivery of protein drugs.

The paper is well organised, the work  appears to be appropriately and effectively executed and a useful presentation of the results and a discussion of their importance is provided.

A few minor suggestions that might further improve the manuscript are offered for consideration by the authors:

Page 2 lines 93-94:  was there any rationale for choosing the model proteins?  Is it simply that they are readily available in quantities  relevant to the study?

Page 3 lines 100-101:  authors comment that lactose and calcium hydrogen phosphate are "less plastic" ( than microcrystalline cellulose), but would they  be appropriately described as exhibiting dominantly brittle fracture compaction mechanism? This may be important to discussing the comparative compaction behaviour, as on application of compression force brittle excipients can generate new surfaces for bonding to potentially produce adequately strong tablets.  In the case of MCC we have plastically deforming excipient with presumably plastically  deforming drug....might  be less desirable combination?

Page 3 line 134:   authors note that BSA was "gently ground".  Was this to effect size reduction, if so what change in particle size was achieved?   Or was this simply to de-aggregate the powder to  assure  mixing behaviour?  Lysozyme was not treated in this way?   Why? Please add appropriate comment to the manuscript.

Page 4, section 2.6:  punches and dies were unlubricated?  No picking, sticking or filming was observed?  Please add comment to the manuscript.

Page 9 towards the end of Discussion section: as the two proteins used have different particle size initially, which may affect compaction behaviour (especially as they are amorphous solids likely to exhibit predominantly  plastic deformation under applied compression force) is there anything in their comparative compaction behaviour to bring out that might be associated with this difference?

Author Response

Dear Reviewer,

Thank you for handling our manuscript extremely fast. We highly value the time spent reviewing and editing our work. Please find our replies to your comments below and the revised manuscript attached.

Reviewer 2 Report

Comments and Suggestions for Authors

The authors reported a binary mixture between two model proteins (i.e. lysozyme and bovine serum albumin,) as compacted excipients up to 30% (w/w) of particulate protein. The authors studied the physical characteristics of the tablets by comparing the mechanical characteristics with the ones of conventional, class tablets. The manuscript  is organized in the way to prove the expected outcome.

The following comments/suggestions are meant to increase the quality of the manuscript:

- the authors reported that they have used a compacting simulator which may lead to the idea that they have used a software but no details were forund; the authors are requested to provide details on the  compacting simulator.

- the authors simulating the compacting the two proteins but no details were inserted concerning the way each protein is compacting by itself, the authors are requested to provide details on this aspect.

-the authors stated:"For developing biological oral tablet products not only the particulate properties, but also protein stability and activity must be in focus.  " however, the authors did not probide any information on the activity before and after compacting; additionally, the stability before and after compacting. The authors are requested to provide details and/or disclaimer on the reason on not probiding such important information. 

Author Response

(The authors gave the same response as above.)

Reviewer 3 Report

Comments and Suggestions for Authors

Dear authors,

The manuscript is well organized and structured, however before publication I would like to elucidate some details:

- Why you choose the concentration of 30% of proteins?

- Which were the criteria to choose the excipients?

- I recommend to add some figures to the manuscript;

- I would like to know the reasons for the diference in the tensile strenghts.

Author Response

(The authors gave the same response as above.)

Round 2

Reviewer 2 Report

Comments and Suggestions for Authors

The authors have answered the addressed queries and updated the document

Author Response

Dear Reviewer,
Thanks for your fast and positive feedback to our manuscript.

Regards,

Reviewer 3 Report

Comments and Suggestions for Authors

Dear Authors,

I recommend the publication of the manuscript in the present form, having in account that all the my comments were well answered.

Congratulations,

Regards

Author Response

(The authors gave the same response as above.)
